# Early Serum Markers for Immune Checkpoint Inhibitor Induced Hypophysitis in Melanoma Patients

**DOI:** 10.3390/cancers16071340

**Published:** 2024-03-29

**Authors:** Fouad Mitri, Devayani Machiraju, Christina Naoum, Jessica C. Hassel

**Affiliations:** 1Heidelberg University, Medical Faculty Heidelberg, Department of Dermatology and National Center for Tumor Diseases (NCT), NCT Heidelberg, a Partnership between DKFZ and University Hospital Heidelberg, 69120 Heidelberg, Germanydevayani.machiraju@med.uni-heidelberg.de (D.M.); christina.naoum@med.uni-heidelberg.de (C.N.); 2Faculty of Biosciences, Heidelberg University, 69120 Heidelberg, Germany

**Keywords:** hypophysitis, immune checkpoint inhibitors, biomarkers, thyroid hormones, hyponatremia

## Abstract

**Simple Summary:**

The effects of immune checkpoint inhibitor (ICI) therapy on 40 melanoma patients who developed immune-related hypophysitis were examined in this study. These patients were compared to 40 matched ones who did not develop hypophysitis. The onset of hypophysitis varied depending on the type of ICI used. Symptoms commonly included fatigue, headaches, and digestive problems. Patients with hypophysitis had low T4 hormone levels regardless of ICI type. However, low T3 and TSH hormone levels were only observed in patients who developed hypophysitis from ipilimumab. A rapid drop in blood sodium levels was also noted at the time of hypophysitis diagnosis. Additionally, the number of eosinophils and lymphocytes in the blood increased consistently in hypophysitis patients. Early diagnosis of hypophysitis is important because of its potential complications. Monitoring patients for specific symptoms and blood value changes could aid in early detection, allowing for prompt intervention and management.

**Abstract:**

Background: Immune checkpoint inhibitors (ICIs) have shown promising anti-tumor activities and are widely used for the treatment of advanced cancers. However, they may lead to immune-related adverse events (irAEs) and some of them, such as hypophysitis, can be life-threatening. Here, early diagnosis is critical. Methods: We retrospectively analyzed 40 melanoma patients who developed hypophysitis during ICI treatment with either ipilimumab and/or anti-PD1 therapy and compared them to 40 control patients who did not develop hypophysitis during the ICI treatment, matched for age, gender, type of immunotherapy, and stage. Clinical data and blood values such as LDH, CRP, TSH, T3, T4, and absolute immune cell counts were retrieved from the medical records. Patient characteristics, laboratory values, progression-free survival, and overall survival were compared between the two groups. Results: Patients with ir-hypophysitis had a median age of 59 years, and most of them were male. Clinically, frequent symptoms were fatigue, headache, dizziness, and gastrointestinal symptoms such as nausea or abdominal pain. The onset of ir-hypophysitis differed much between ipilimumab- (median 8 weeks) and anti-PD1 (median 40 weeks)-induced hypophysitis (*p* < 0.001). At baseline, besides a slightly increased CRP level (*p* = 0.06), no differences were observed in patients who later developed hypophysitis compared to the control. After treatment started, hypophysitis patients showed a constant and significant decline in T4 levels from the start of therapy until diagnosis (*p* < 0.05), independent of the ICI treatment regime. However, a decline in T3 and TSH was only noted in patients with ipilimumab-induced ir-hypophysitis. Furthermore, serum sodium levels declined rapidly at the diagnosis of hypophysitis (*p* < 0.001). In addition, there was a constant increase in the absolute counts of eosinophils and lymphocytes from baseline in hypophysitis patients (*p* < 0.05). Conclusion: Ir-hypophysitis reveals different clinical pictures and onset times depending on the ICI regime used. Whereas a drop in T4 levels was indicative of developing hypophysitis independent of the ICI regime, TSH levels only declined in patients under ipilimumab-based ICI regimes. To best monitor our patients, it is important to recognize these differences.

## 1. Introduction

Melanoma remains a potentially fatal malignancy, with an increasing incidence [1]. Treatments with ICIs, including antibodies against cytotoxic T-lymphocyte antigen 4 (CTLA-4) and programmed cell death-1 (PD-1), activate the immune system and have shown anti-tumor effects with increasing survival rates in advanced melanoma patients [2]. However, ICI-mediated immune activation is not tumor-specific; thus, treatment with ICI can initiate an immune response against normal tissue. The resulting events are called immune-related adverse events (irAE) [3]. The latter can affect many organs and cause endocrinopathies, including hypophysitis.

Hypophysitis is an inflammation of the pituitary gland. Its etiology varies, ranging from primary (idiopathic) to secondary (known). ICI-related hypophysitis has been mainly described after CTLA-4 exposure; however, other ICIs have also been implicated [4]. The reported incidence of this irAE varies between ≤1 and 10% [5,6]. Combining different checkpoint inhibitors increases the incidence of endocrine adverse events [7].

The exact pathogenesis of ICI-related hypophysitis is unknown. There is ongoing debate regarding the incidence of the disease, the risk factors, the pathways involved, and the most effective strategies for diagnosing and monitoring patients [8]. However, it has been hypothesized that anti-CTLA-4 initiates an autoimmune process that targets unidentified pituitary antigens. Thyrotrophs and lactotrophs in the pituitary express CTLA-4, suggesting that type II hypersensitivity is associated with the effect of anti-CTLA-4 antibodies in the pituitary gland [5,9,10]. A severe complication of this adverse event is the acute (secondary) adrenal crisis with hypotension, fatigue, and a sepsis-like presentation [11]. Regarding medication, glucocorticoids are frequently used to treat patients with ICI-associated hypophysitis. ICIs can be resumed after the acute disease subsides [8]. Almost 86–100% of affected patients require long-term hydrocortisone replacement therapy [12].

The diagnosis of hypophysitis is based on clinical as well as laboratory findings. In radiology, MRI frequently reveals diffuse enlargement and/or enhancement of the pituitary gland, but because up to 25% of scans may seem normal, it should not be used to rule out hypophysitis [13]. Besides, PD-1-induced cases usually have no radiological changes on MRI. Lab findings include hyponatremia, which is due to acute secondary adrenal insufficiency and is more frequently found in ICI-related hypoadrenalism in comparison to other etiologies [14]. Apart from hyponatremia, patients with ICI-induced hypophysitis usually present with other endocrine deficiencies, for example, in the thyroid stimulating hormone (TSH) (84%), adrenocorticotropic hormone (ACTH) (80%), and gonadotropin (76%) [15,16]. Other laboratory parameters showing changes in adrenal insufficiency after ICI therapy include increased leucocytes, a relative neutrophil count, a decreased lymphocyte count, and increased eosinophil counts [6,17,18,19,20].

In general, diagnosing ICI-related hypophysitis can be difficult since the presenting symptoms are usually unspecific: fatigue, loss of weight, and anorexia, which are symptoms often associated with the underlying malignancy. Thus, immune-related pituitary insufficiencies could be overlooked when pituitary hormones are not measured regularly. Furthermore, in some cases, depending on the timing or severity of clinical presentation, patients do not have the opportunity for a more formal assessment of the hypothalamic-pituitary-adrenal axis before the need for urgent clinical treatment. Notably, late detection of hypophysitis, mainly associated with adrenal insufficiency, is associated with high morbidity and mortality [3]. Thus, better prognostic factors and possibly early markers for this disease are required. Therefore, this study aimed to investigate early biomarkers in the blood of melanoma patients who developed hypophysitis during ICI treatment.

## 2. Methods

### 2.1. Patient Selection and Clinical Data Collection

This is a single-center retrospective analysis comparing melanoma patients who have been clinically diagnosed with hypophysitis after undergoing immunotherapy with ipilimumab (10 mg/kg; 3 mg/kg) or ipilimumab (3 mg/kg) plus nivolumab (1 mg/kg) or pembrolizumab (200 mg fixed dose) vs. matched control patients who did not develop hypophysitis. We searched the database in the Department of Dermato-Oncology for melanoma stage IV patients who presented to us from November 2010 until May 2020. We reviewed the charts for demographics (age, symptoms, therapy, development of immune-related adverse events, cranial MRI, stage of disease), as well as laboratory parameters (leukocytes, lymphocytes, eosinophils, neutrophils, sodium, TSH, Triiodothyronine (T3), tetraiodothyronine (T4), glucose, CRP, LDH), before the start of immunotherapy, 2–4 weeks before the diagnosis (depending on the availability of data), and at the time of diagnosis of hypophysitis. If there were no laboratory values documented, such as CRP and LDH, at the time of treatment, the values were reported as missing in Table 1.

The timeframe of 2 to 4 weeks was selected to assess values shortly before the onset of the disease, ideally aiming for 4 weeks. However, due to some missing data, a range of 2 to 4 weeks was opted for.

The diagnosis of hypophysitis was made clinically based on typical symptoms and the detection of low ACTH (<10 pg/mL) and morning cortisol levels (<56 ng/mL). Patients with unavailable ACTH and cortisol levels, those who had received glucocorticoid therapy prior to presentation, and those with missing lab values 2 to 4 weeks before the diagnosis were excluded. Thus, 40 out of 63 patients with clinical and laboratory evidence of ir-hypophysitis were selected (Appendix A).

The control group was randomly chosen from our extensive database of hundreds of melanoma patients who either underwent or are undergoing immunotherapy at our oncology center. Initially, a list of patients matching in terms of age, gender, type of immunotherapy, and cancer stage was generated. From this list, 40 control patients were randomly selected using the lottery method. Therefore, for each patient with hypophysitis, one control patient was matched. The laboratory values of an estimated date of 8 weeks after the start of ipilimumab therapy and 40 weeks after PD-1 therapy were then used for measurement. The values 2–4 weeks prior to the extrapolated diagnosis date were also evaluated. The Medical Faculty of Heidelberg’s Ethical Committee approved the retrospective patient data analysis with the code S-454/2015. The principles outlined in the Declaration of Helsinki were followed.

### 2.2. Statistical Analysis

The Chi-square probability distribution test was used to analyze the difference in categorical variables between the control and ir-hypophysitis groups, whereas the MWU test was performed to analyze the difference in continuous variables between the groups. Changes in blood parameters from repeated measurements on the same individuals within the group (i.e., at different time points: baseline vs. 3 weeks before diagnosis vs. the time of diagnosis) were analyzed using the Wilcoxon-rank test (WCR). In addition, ROC curve analysis was performed to find the early biomarkers using hypophysitis as an event. The time for hypophysitis development was defined as the time from the start of immunotherapy until the clinical diagnosis of hypophysitis. Progression-free survival (PFS) was defined as the interval from the commencement of immunotherapy to the date of disease progression or death from any cause. Overall survival (OS) was defined as the interval from the commencement of immunotherapy until death, and patients who were still alive were censored at the date of the last contact. Kaplan-Meier analysis and the log-rank test were employed to assess survival disparities between the control and ir-hypophysitis groups. Statistical analyses were conducted using SPSS version 29 (IBM), while GraphPad Prism version 8 (GraphPad Software, Inc., La Jolla, CA, USA) was utilized for data visualization and graph generation. Column graphs present median values and 95% confidence intervals (CI) using bars and lines. Two-sided *p*-values were calculated, with significance set at *p* < 0.05, indicating a statistically significant difference.

## 3. Results

### 3.1. Patient Characteristics

Eighty patients with advanced melanoma (stage III or IV) receiving ICI treatment were included in the study. Among them, forty patients had developed ir-hypophysitis during the ICI treatment, and forty patients with matching clinical characteristics who did not develop hypophysitis were included as a control group (Appendix A). Baseline characteristics and clinical outcomes are described in Table 1. Patients with ir-hypophysitis had a median age of 59 years, 68% of whom were male patients. Clinical symptoms included mainly fatigue, dizziness, and weakness. In addition to matched clinical characteristics (age, gender, type of ICI), there were no significant differences in non-matched baseline patient characteristics, such as other irAE and serum lactate dehydrogenase (LDH) levels at baseline, between the two groups. However, the CRP values were trending higher at baseline in patients who developed hypophysitis than in the control group (*p* = 0.06).

### 3.2. Clinical Characteristics of Ir-hypophysitis

First, we analyzed the time to hypophysitis development between patients treated with different ICI regimens (Figure 1a). Patients who received ipilimumab alone or in combination with nivolumab developed hypophysitis after a median of 8 weeks and therefore faster than patients who received anti-PD1 monotherapy, who developed the ir-hypophysitis after a median of 40 weeks (*p* < 0.001). Regarding the symptoms, they were similar mostly in nature in both groups, i.e., both had weakness, dizziness, hypotension, altered mental status, and/or gastrointestinal symptoms such as abdominal pain or nausea. However, cephalgia was only seen in the ipilimumab group in 13/31 (43.3%) vs. none in the PD-1 group.

Twenty patients (50%) with ir-hypophysitis also experienced other ir-AEs, including rash (6), arthritis (6), colitis (5), hepatitis (4), pancreatitis (3), thyroiditis (3), and other (3). Among them, ir-hypophysitis was the first ir-AE to be observed in six patients, with one patient having both hypophysitis and hepatitis at the same time. In the other fourteen patients, other ir-AEs were detected before the diagnosis of hypophysitis, with rash being the most frequent (5), followed by arthritis (3), colitis (2), thyroiditis (2), and others (2). In the two patients who developed colitis as the first ir-AE, the treatment had to be discontinued, and hypophysitis was diagnosed afterward. Detailed information on other ir-AEs and their chronological sequence has been illustrated in Appendix A.

In terms of clinical efficacy, 25 (63%) patients with ir-hypophysitis responded to ICI therapy, compared to 22 (58%) in the control group (*p* = 0.644). Median PFS was 7 months (95% CI: 15–36) in the control group and 17 months (95% CIs: 19–41) in the ir-hypophysitis group. There was no significant difference in PFS (*p* = 0.376; Figure 1b) or OS (*p* = 0.355; Figure 1c), keeping in mind the small number of included patients for such analysis.

### 3.3. Laboratory Values Preceding and at the Time of Diagnosis of Ir-hypophysitis

Comparison of the laboratory values at the time of diagnosis of ir-hypophysitis with matched controls revealed that patients with ir-hypophysitis had significantly lower levels of serum sodium levels (as secondary phenomena to low ACTH levels; *p* < 0.001), as well as TSH (*p* = 0.027) and fT4 (*p* < 0.001) but not fT3 (*p* = 0.094; Figure 2). Additionally, we noticed a significant increase in peripheral blood lymphocytes (*p* = 0.008) and eosinophils (*p* = 0.021) at the time of diagnosis (Figure 2). Besides, similar to the baseline, elevated CRP at the time of diagnosis was more likely seen in ir-hypophysitis patients compared to the controls (*p* = 0.096).

To search for early markers for ir-hypophysitis in the peripheral blood, we analyzed the respective laboratory values right before the initiation of ICI treatment and 3 weeks before the diagnosis of the irAEs (Figure 3). Here, a significant reduction in T4 hormone levels was detected already around 3 weeks before the diagnosis in ir-hypophysitis patients when compared to the patients in the control group (*p* = 0.001). Although the same trend was observed with TSH levels, the difference was only significant at the time of the ir-hypophysitis diagnosis (*p* = 0.027). Besides, we noticed that T3 levels were constantly reducing, but no difference could be seen between the control and ir-hypophysitis groups. Accordingly, T4 levels 3 weeks before diagnosis were more significant in predicting ir-hypophysitis than both TSH and T3 values (*p* = 0.002; AUC: 0.713; 95%CI: 0.595–0.832; Appendix A). Moreover, we have also seen an increase in peripheral blood lymphocytes and eosinophils in patients developing ir-hypophysitis 3 weeks before the diagnosis (*p* = 0.004 and *p* = 0.008, respectively) that remained until the diagnosis (*p* = 0.03 and *p* = 0.021) in comparison to the controls (Figure 2).

Interestingly, comparing different ICI regimens, T4 levels were significantly reduced upon both treatment regimens (*p* < 0.05; Figure 3 and Appendix A), whereas a TSH drop could be detected only in ipilimumab-induced ir-hypophysitis (*p* = 0.072). In patients treated with anti-PD1 monotherapy, TSH levels even increased slightly (*p* = 0.051) (Figure 3 and Appendix A). Compared to the control group, all 3 parameters significantly decreased only in ipilimumab-induced ir-hypophysitis; anti-PD1 monotherapy-induced ir-hypophysitis revealed only a significant difference in T4 levels (Appendix A). Concerning serum sodium levels, a significant reduction at the time of ir-hypophysitis was detected in both treatment regimens (*p* < 0.05); however, patients with ipilimumab-induced ir-hypophysitis developed a stronger sodium drop at the time of diagnosis (Figure 3).

## 4. Discussion

Hypophysitis is a known but rare autoimmune toxicity of ICIs. It is not easy to detect, and its diagnosis is usually presumptive. To prevent delay in diagnosis, it is crucial to anticipate its onset. Early diagnosis of ICI-induced hypophysitis may be delayed due to the non-specific symptoms and clinical manifestations, as well as the lack of readily available endocrinologic laboratory examinations in primary care. Furthermore, ICI-related hypophysitis is not always seen radiologically.

In this single-institution study, we investigated the clinical and laboratory characteristics of ICI-induced hypophysitis in melanoma patients treated with either anti-PD1 monotherapy or ipilimumab +/− nivolumab. We observed a significant reduction in T4 hormone levels approximately 3 weeks before the clinical diagnosis of ir-hypophysitis, suggesting its importance in early diagnosis. Similar changes in TSH and sodium levels were observed, however, at the time of the ir-hypophysitis diagnosis. Our study supports that regular monitoring of thyroid hormones will help in the early diagnosis of ir-hypophysitis.

Clinically, as reported in the literature [21], males were more commonly affected than females. A striking difference between ipilimumab- and anti-PD1-induced ir-hypophysitis was the time to clinical onset, with 8 weeks at median for ipilimumab and 40 weeks for anti-PD1. This is especially important as hypophysitis under anti-PD1 therapy occurs not within the first 12 weeks of treatment and might be overlooked first because of a less acute onset. The delayed onset of PD-1 treatment in comparison to CTLA-4 is well described in the literature [4]. The CTLA-4 IgG1 or IgG2 therapeutic antibodies can induce ir-hypophysitis through type II hypersensitivity reactions by directly binding to CTLA-4 on pituitary cells and type IV hypersensitivity reactions by triggering antibody-dependent cell-mediated cytotoxicity [22,23]. In contrast, PD-1 inhibitors are IgG4 antibodies, suggesting that PD-1-triggered ir-hypophysitis may be similar to IgG4-related hypophysitis. Nevertheless, the exact mechanism behind the differences in frequency and time between CTLA-4 and PD-1-induced ir-hypophysitis is yet to be explored.

When it comes to symptoms, both the ipilimumab- and anti-PD1 groups had similar symptoms of weakness, dizziness, hypotension, altered mental status, and/or nausea/abdominal pain. Nevertheless, the ipilimumab group was more distinguished from the PD-1 group by the presence of headaches. This supports prior studies [4,24].

Although we did not find any blood biomarkers indicating ICI-hypophysitis at baseline, using a control cohort, we were able to show that T4 hormone level was the best value to early detect ir-hypophysitis in both ipilimumab- and anti-PD1 induced cases. In addition, patients with ipilimumab-induced ir-hypophysitis revealed a decline in TSH and T3 levels that were not seen in patients under anti-PD1 monotherapy.

When it comes to the thyroid hormones, we did not observe any difference in TSH hormone at baseline, and although the TSH went down around three weeks before diagnosis in hypophysitis, a significant difference when compared to the control group was only observed at the time of diagnosis. Meanwhile, we observed a strong and significant reduction in T4 hormone levels around 2–4 weeks before the clinical manifestation of ICI-induced hypophysitis, suggesting the importance of regular thyroid hormone measurements, especially T4 along with TSH. In a previous study, a fall in fT4 before treatment cycles 3 and 4 proved to be better at predicting the subsequent development of hypophysitis than TSH, whose fall was not statistically significant [25].

Blood sodium levels showed a dramatic drop at the time the hypophysitis was diagnosed; however, in ipilimumab-treated patients, in contrast to anti-PD1 monotherapy, this was more acute, without detectable first signs of reduction 3 weeks before the diagnosis. This means, on the other hand, that in anti-PD1 monotherapy-induced ir-hypophysitis, a reduction in serum sodium might be an early marker for ir-hypophysitis.

The primary mechanism for developing hyponatremia is the decreased capacity of free water excretion due to elevated antidiuretic hormone levels. However, ir-hypophysitis usually affects the adenohypophysis and not the posterior pituitary lobe, where the antidiuretic hormone is secreted. Hence, this is probably a bystander effect of the inflammation. Regardless of the etiology and magnitude of electrolyte imbalance, hyponatremia was associated with higher mortality risk, more extended hospital stays, and lower PFS in patients with cancer [1,2]. Interestingly, patients developing ir-hypophysitis had better overall survival than malignant melanoma and non-small cell lung carcinoma patients [26]. However, as in our cohort, recent analysis did not further support this. Accordingly, a previous report in melanoma patients treated with ipilimumab also reported no association between ir-hypophysitis and survival [27]. However, the relatively small sample size and the study design may account for such discrepancies and should be kept in mind.

Regarding further early markers of hypophysitis, Sekizaki et al. suggest that ACTH elevation may reflect the destruction of the pituitary gland, hypothesizing that this finding may be necessary for the early detection of ICI-induced hypophysitis [28]. ACTH is produced by the pituitary gland and triggers the production of cortisol from the adrenal cortex cells. Impaired ACTH release is very frequent in the early stages of primary hypophysitis; hence, fluctuations in ACTH in the blood may indicate the development of ir-hypophysitis [29]. However, we did not measure it in our cohort as ACTH levels are not part of the clinical routine for patients treated with immune checkpoint blockers. Interestingly, a study showed that some patients with ICI-hypophysitis experienced a temporary increase in ACTH levels prior to developing secondary adrenal insufficiency; thus, tracking changes in ACTH levels may aid in anticipating the onset of the disease [26].

The findings mentioned above may hold clinical significance for patient monitoring; however, it is crucial to acknowledge several limitations of our study. These include its retrospective nature and the single institutional setting with a limited observational period. In addition, the sample size was relatively small, and the control group was randomly selected from a large list of patients treated with ICI to match with the ir-hypophysitis cohort based on a few similar clinical features, namely age, gender, type of immunotherapy, and cancer stage. There may be differences in other clinical characteristics between the two groups, which could introduce a selection bias, especially in the case of small group sizes. A larger control group could have been beneficial. Therefore, further prospective studies are necessary to validate our observations. These studies should involve larger patient cohorts and explore biomarkers associated with the efficacy and safety of immune checkpoint inhibitors.

## 5. Conclusions

Since ICIs have become a mainstream therapy for many cancers, monitoring for symptoms and laboratory changes in autoimmune hypophysitis is required to diagnose this condition early for prompt management. Our study highlights the importance of regularly monitoring the T4 hormone, whereas TSH was only helpful in patients with ipilimumab ICI regimes. Hyponatremia was a frequent finding in patients with ir-hypophysitis and therefore might be indicative and lead to further work-up, including ACTH and cortisol. A broader once-immuno-endocrinologic approach would improve patient care even more, and therefore further research to identify better markers to predict ir-hypophysitis is paramount.

## Figures and Tables

**Figure 1 cancers-16-01340-f001:**
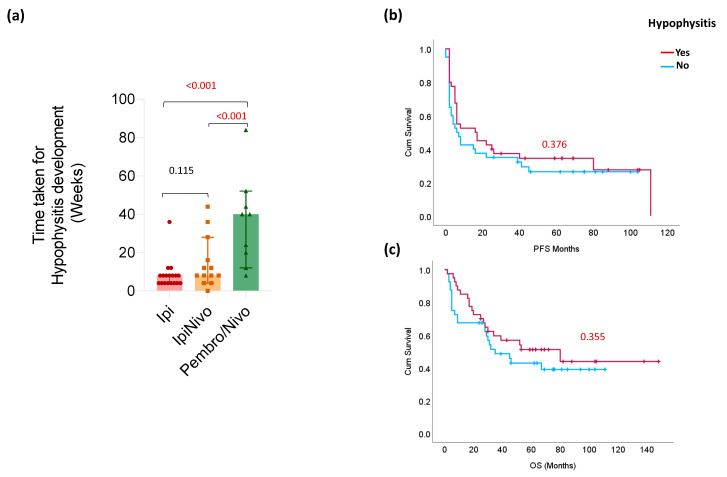
(**a**) Bar chart representing the time taken for hypophysitis development in different ICI regimes. The lines represent the median and 95% CI for all charts, respectively. *p* values are presented above the respective group on top of the chart. (**b**,**c**) Kaplan Meier curves for PFS (**b**) and OS (**c**) according to hypophysitis (red) or no hypophysitis (blue) groups. *p*-values refer to the log-rank test.

**Figure 2 cancers-16-01340-f002:**
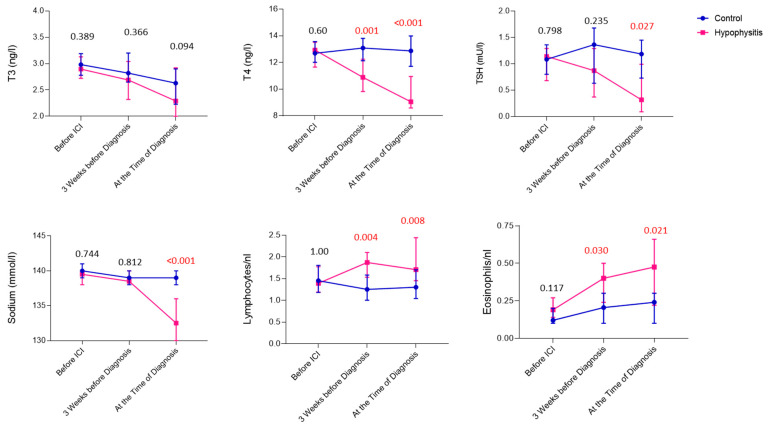
Comparison of blood parameters between control (blue) and ir-hypophysitis patients (pink). The lines represent the median and 95% CI for all charts, respectively. *p*-values are presented above the respective group on top of the chart; a *p*-value less than 0.05 indicates a significant difference between the groups.

**Figure 3 cancers-16-01340-f003:**
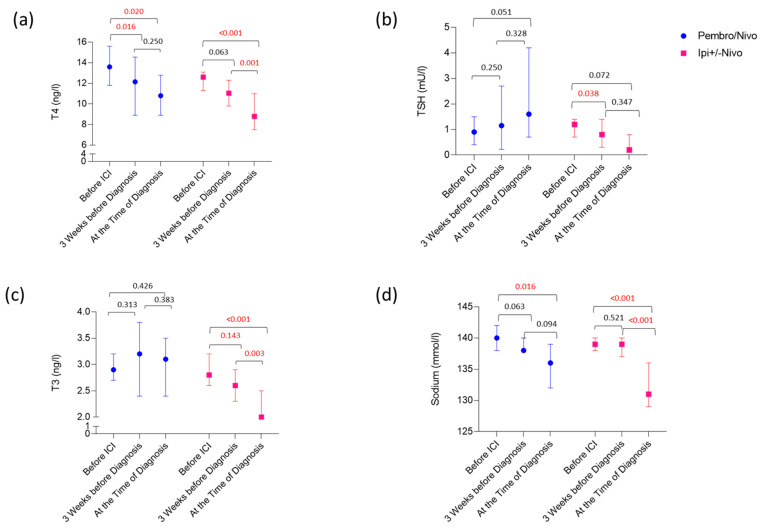
Influence of ICI regimes on T4 (**a**), TSH (**b**), T3 (**c**) and sodium levels (**d**) in ir-hypophysitis patients. The lines represent the median and 95% CI for all charts, respectively. *p* values are presented above the respective group on top of the chart. *p*-values less than 0.05 indicate a significant difference between the respective time points.

**Table 1 cancers-16-01340-t001:** Patient characteristics of advanced melanoma patients.

	Control (40)	Ir-Hypophysitis (40)	*p*-Value
**Age (Years)**			
Median (Range)	59 (17–77)	59 (20–77)	0.8
	** *n* ** **(%)**	** *n* ** **(%)**	
**Gender**			1.0
Male	27 (68)	27 (68)	
Female	13 (32)	13 (32)	
**Type of ICI**			1.0
Pembro/Nivo	9 (23)	9 (23)	
Ipi+/−Nivo	31 (77)	31 (77)	
**Other irAEs**			1.0
Yes	20 (50)	20 (50)	
No	20 (50)	20 (50)	
**CRP at Baseline**			0.06
Normal	19 (48)	27 (68)	
Elevated	20 (50)	11 (28)	
Missing	1 (2)	2 (4)	
**LDH at Baseline**			1.0
Normal	33 (83)	33 (83)	
Elevated	7 (17)	6 (15)	
Missing		1 (2)	
**PFS (Months)**			0.2
Median (95%CI)	7 (15–36)	17 (19–41)	
**OS (Months)**			0.3
Median (95%CI)	32 (30–51)	48 (38–60)	

## Data Availability

Data are contained within the article and Appendix A.

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
