# Peer review of "Early Serum Markers for Immune Checkpoint Inhibitor Induced Hypophysitis in Melanoma Patients"

_cancers, 2024, doi:10.3390/cancers16071340_

Round 1

Reviewer 1 Report

Comments and Suggestions for Authors

This is an interesting clinical study on the management of hypophysitis following administration of therapeutic monoclonal antibodies to patients with melanoma cancers. The background , research problem and clinical need for the study are clearly explained and the methods seem standard. The authors worked their way to providing some early markers that precede the development of the disease that may be useful for other clinicians. The limitations of the study are fairly discussed. 

Author Response

Thank you for reviewing this article, we really appreciate your feedback. 

Cordially, 

Reviewer 2 Report

Comments and Suggestions for Authors

The reviewed scientific paper investigates immune checkpoint inhibitor (ICI)-induced hypophysitis in melanoma patients, emphasizing the importance of early detection. The study provides comprehensive insights into clinical and laboratory characteristics, highlighting T4 levels as a potential early marker. While acknowledging limitations, the discussion suggests regular monitoring of T4 and broader immuno-endocrinologic approaches to enhance patient care, with a call for further prospective studies. I commend the authors for their thorough analysis and transparent acknowledgment of study limitations. Some comments are listed below for authors consideration:

Introduction:

1. While the introduction outlines the background well, it could explicitly state the study's objectives or research questions. Adding a clear statement about the aim of investigating early biomarkers in melanoma patients with ICI-induced hypophysitis would enhance focus.

2. The section discussing the pathogenesis of ICI-related hypophysitis is informative, but a bit more detail on the current understanding or controversies in this area could enrich the introduction.

Methods:

  1. Explanation of Time Points: The rationale for choosing specific time points for data collection (2-4 weeks before diagnosis, etc.) could be briefly discussed to provide context for readers.
  2. Additional Information on Matching: While the control group is mentioned to be matched for age, gender, immunotherapy, and stage, additional details on the matching process or any potential challenges encountered during matching would be beneficial.

Results:

    1. Discussion of Clinical Outcomes: While the study mentions clinical outcomes, such as response rates to ICI therapy and PFS/OS, a brief discussion or interpretation of these findings in the context of existing literature could be beneficial.
    2. Discussion of Limitations: A brief section discussing the limitations of the study, such as the retrospective nature or the relatively small sample size, would add transparency to the research.
    3. Further Details on Statistical Analysis: While the statistical methods are mentioned, providing a brief rationale for the choice of specific statistical tests and more context for the interpretation of p-values could enhance the readers' understanding.
    4. Explanation of Missing Values: There are instances of missing values in the dataset (e.g., CRP values at baseline). Providing a brief explanation or discussing potential implications would be useful.

Discussion:

  1. ACTH Elevation: The discussion mentions that ACTH elevation may be a potential early marker but was not measured in the study. A brief exploration of the potential significance of ACTH in the context of hypophysitis could be valuable for readers.

4. Clarification on the Relationship Between Hyponatremia and Mortality: The section mentions the association of hyponatremia with higher mortality risk but notes that recent analysis did not support this in the cohort. A brief discussion on the potential reasons for this discrepancy or the need for further research would be insightful

Author Response

Answers to the reviewers:

Reviewer 2:

Dear reviewer, thank you for taking your time and for giving us valuable input. 

Comments of the reviewer

Answer

1. While the introduction outlines the background well, it could explicitly state the study's objectives or research questions. Adding a clear statement about the aim of investigating early biomarkers in melanoma patients with ICI-induced hypophysitis would enhance focus.

This study aimed to investigate early biomarkers in the blood of melanoma patients receiving ICI to diagnose hypophysitis before clinical onset.

We highlighted the focus of the study in the introduction part (Page:2; Line: 85-86).

2. The section discussing the pathogenesis of ICI-related hypophysitis is informative, but a bit more detail on the current understanding or controversies in this area could enrich the introduction.

A sentence has now been added to mention the controversies in the field (Page:2; Line: 52-56).

Explanation of Time Points: The rationale for choosing specific time points for data collection (2-4 weeks before diagnosis, etc.) could be briefly discussed to provide context for readers.

The period of 2 to 4 weeks was chosen to evaluate values just prior to the clinical onset of hypophysitis. Since patients undergo immunotherapy at intervals of 2 to 4 weeks depending on the specific substance, this aligns with the frequency of blood testing for these patients.

The selection of the timeframe was explained in the revised manuscript (Page: 3; Line: 100-105)

Additional Information on Matching: While the control group is mentioned to be matched for age, gender, immunotherapy, and stage, additional details on the matching process or any potential challenges encountered during matching would be beneficial.

The control group was randomly chosen from our extensive database of hundreds of melanoma patients who either underwent or are undergoing immunotherapy at our oncology center. Initially, a list of patients matching in terms of age, gender, type of immunotherapy, and cancer stage was generated. From this list, 40 control patients were randomly selected using a lottery method. Therefore, for each patient with hypophysitis, one control patient was matched.

A clarification of the matching process has been added in the manuscript (Page: 3; Line: 112-117).

Discussion of Clinical Outcomes: While the study mentions clinical outcomes, such as response rates to ICI therapy and PFS/OS, a brief discussion or interpretation of these findings in the context of existing literature could be beneficial.

In our study we did not find significant differences in response rates, PFS, and OS, however, this might just be based on the small number of patients for these efficacy outcomes.

In the literature, results differ, with studies showing a better clinical outcome and studies that do not.  We added this in the revised manuscript (Pages: 8-9; Lines 276-281)

Discussion of Limitations: A brief section discussing the limitations of the study, such as the retrospective nature or the relatively small sample size, would add transparency to the research.

We added the limitations of our study at the end of the discussion (Page 9; Line: 301-309)

Further Details on Statistical Analysis: While the statistical methods are mentioned, providing a brief rationale for the choice of specific statistical tests and more context for the interpretation of p-values could enhance the readers' understanding.

The details on the choice of specific statistical tests and the interpretation of p-values were elaborated in the revised manuscript (Page: 3; Line: 124-142)

Explanation of Missing Values: There are instances of missing values in the dataset (e.g., CRP values at baseline). Providing a brief explanation or discussing potential implications would be useful.

Missing values are rare in our study; however, it happened in the clinical routine that special lab values (e.g. CRP, LDH) were not tested at a time point and were then indicated as missing in Table 1. We added a sentence to explain this in the revised manuscript (Page: 3; Line 104-105)

ACTH Elevation: The discussion mentions that ACTH elevation may be a potential early marker but was not measured in the study. A brief exploration of the potential significance of ACTH in the context of hypophysitis could be valuable for readers.

Yes, there are some reports of transient ACTH increases most likely because of pituitary gland destruction. In our study, we did not report it as we do not regularly measure ACTH and cortisol in the monitoring of the patients but test it only when hypophysitis is suspected.

However, we thank you for the valuable comment on ACTH and integrated discussion on this. (Page: 9; Line: 282-292)

Clarification on the Relationship Between Hyponatremia and Mortality: The section mentions the association of hyponatremia with higher mortality risk but notes that recent analysis did not support this in the cohort. A brief discussion on the potential reasons for this discrepancy or the need for further research would be insightful

When it comes to hyponatremia and survival, we mentioned that in some reports patients treated with ipilimumab also reported no association between ir-hypophysitis and survival and that our relatively small sample size and the study design may account for such discrepancies (Page: 9; Line: 273-277) 

Thank you for the important and constructive feedback.

Reviewer 3 Report

Comments and Suggestions for Authors

The authors of the article point out that early serum markers of immune checkpoint inhibitor-induced hypophysitis in melanoma patients have very high clinical significance. I strongly agree with the author’s opinion and this article is informative and should be published without any revisions.

1. What is the main question addressed by the research?

The authors of the article point out that early serum markers of immune checkpoint inhibitor-induced hypophysitis in melanoma patients have very high clinical significance. I strongly agree with the author’s opinion and this article is informative and should be published without any revisions.

2. Do you consider the topic original or relevant in the field?

I think that the topic is relevant in the field.

3. What specific improvements should the authors consider regarding the

methodology? What further controls should be considered?

No specific improvements and controls should be considered.

4. Are the conclusions consistent with the evidence and arguments presented

and do they address the main question posed?

Yes, the conclusions consistent with the evidence and arguments presented

and they address the main question posed.

5. Are the references appropriate?

References and others were not found to be inappropriate

6. Please include any additional comments on the tables and figures.

No comments

Author Response

Dear reviewer,

Thank you for your constructive and important feedback and that you took the time to provide us with the comments. 

Cordially, 

Reviewer 4 Report

Comments and Suggestions for Authors

This is an interesting study for the still little-known topic of the side effects of immunotherapy drugs in melanoma patients. Therefore it is a study worthy of constructive critical attention. A few points need to be clarified: how were the control patients recruited? The modality used is not acceptable, it is not possible to superselect patients to be compared as has been done. It would have been better to maximize the number of control patients, i.e. to use all possible data observable in all patients you have. What is the sense of selection of 40 versus 40 for? It is not a randomized trial but a research of toxicity markers..

However, the study, even if conceptually simple, flows well. The research aspects should be addressed more deeply: 1) how many patients have discontinued immunotherapy and whether these have seen hypophysitis regress, 2) what other toxicities have patients had? Just hypophysitis? Because you haven't assessed the prevalence of pituitary antibodies. Pituitary glands indeed expressed CTLA-4 at both RNA and protein levels, particularly in a subset of prolactin- and thyrotropin-secreting cells.

In the calculation of OS and PFS it is not clear whether patients have always continued treatments or whether due to toxicity they have been interrupted and even then eventually resumed. I would advise the authors to be a little more courageous in the discussion by adding research hypotheses and not just saying "new studies are needed". It would be interesting to know or discuss why IPI is toxic after 8 weeks and anti-PD1 after many more weeks

Reference 25 is incomplete

Comments on the Quality of English Language

The English language is well written, correct, a little easy for an article to be presented in a high-prestige medical journal. It could also be accepted but I would recommend a quick review by a native speaker

Author Response

Reviewer 4:

Dear reviewer, thank you for taking your time and for giving us valuable input. 

We were able to take many of your remarks into consideration.

Comments of the reviewer

Answer

A few points need to be clarified: how were the control patients recruited? The modality used is not acceptable, it is not possible to superselect patients to be compared as has been done. It would have been better to maximize the number of control patients, i.e. to use all possible data observable in all patients you have. What is the sense of selection of 40 versus 40 for? It is not a randomized trial but a research of toxicity markers.

The control group was randomly chosen from our database of hundreds of melanoma patients who either underwent or are undergoing immunotherapy at our oncology center. Initially, a list of patients matching in terms of age, gender, type of immunotherapy, and cancer stage was generated. From this list, 40 control patients were randomly selected using a lottery method (Page: 3; Line: 111-116).

We aimed to have a comparable cohort size of patients with and without ir-hypophysitis. However, we are aware that more clinical markers might be relevant, the reason why patients were randomly selected.

Nevertheless, we also now mentioned this in our limitations in the revised manuscript (Page:9; Line 299-307)

How many patients have discontinued immunotherapy and whether these have seen hypophysitis regress,

We thank the reviewer for the comment; however, this point could not be elaborated in time.

What other toxicities have patients had? Just hypophysitis?

Eighteen patients (45%) with ir-hypophysitis also experienced other irAEs such as colitis, arthritis, pancreatitis, etc. We have included this information in our revised manuscript (Page: 5; Line: 165-166).

Because you haven't assessed the prevalence of pituitary antibodies. Pituitary glands indeed expressed CTLA-4 at both RNA and protein levels, particularly in a subset of prolactin- and thyrotropin-secreting cells.

We agree with Reviewer#4, measuring the antibodies would've been very interesting. However, as we are reporting on routine laboratory parameters, we do not have this information.

Of note, we added a discussion on the pathogenesis (Page: 8; Line: 238-245).

In the calculation of OS and PFS it is not clear whether patients have always continued treatments or whether due to toxicity they have been interrupted and even then, eventually resumed.

PFS and OS data were collected irrespective of ir-hypophysitis knowledge.

In our ir-hypophysitis cohort,19/40 patients discontinued the treatment due to side effects, however, most of them were also diagnosed with other irAEs that led to treatment discontinuation. We did not analyze survival in these subgroups as the cohort is small already and the focus of the manuscript is on biomarkers for hypophysitis.

But we added this information for the reader in the description of the patients.

I would advise the authors to be a little more courageous in the discussion by adding research hypotheses and not just saying "new studies are needed". It would be interesting to know or discuss why IPI is toxic after 8 weeks and anti-PD1 after many more weeks.

We completely restructured the discussion and added more information and hypotheses why there might be a difference in the ir-hypophysitis of CTLA-4 inhibitors vs PD-1 (Page 8; Line: 238-245)

Reference 25 is incomplete.

Detailed information of reference 25 has now been provided in the revised manuscript (12: Line: 440-443)

Thank you for the important and constructive feedback.

Round 2

Reviewer 4 Report

Comments and Suggestions for Authors

At least 2 point to be improved:

1)Patient selection and pseudo-selection are very unusual

2)"Eighteen patients (45%) with ir-hypophysitis also experienced other adverse events such as colitis, arthritis, pancreatitis" Still too vague, undefined, unscientific. Which toxicity appeared first? If colon toxicity appeared first, was treatment stopped? How many patients have had all or some toxicities? And which ones? What is the sequence of toxicity?

Pls specify better

Author Response

Please find attached the answer to the comments. 

Thank you 
